# Controversies and Opportunities in the Clinical Daily Use of the 21-Gene Assay for Prognostication and Prediction of Chemotherapy Benefit in HR+/HER2- Early Breast Cancer

**DOI:** 10.3390/cancers15010148

**Published:** 2022-12-27

**Authors:** Flavia Jacobs, Mariangela Gaudio, Chiara Benvenuti, Rita De Sanctis, Armando Santoro, Alberto Zambelli

**Affiliations:** 1IRCCS Humanitas Research Hospital, Humanitas Cancer Center, Via Manzoni 56, 20089 Rozzano, MI, Italy; 2Department of Biomedical Sciences, Humanitas University, Via Rita Levi Montalcini 4, 20090 Pieve Emanuele, MI, Italy; 3Academic Trials Promoting Team, Institut Jules Bordet, L’Université Libre de Bruxelles (U.L.B.), 1070 Bruxelles, Belgium

**Keywords:** genomic signature, OncotypeDX, precision medicine, HR-positive early breast cancer, adjuvant chemotherapy, clinicopathological factors, node-positive, premenopausal, special histologies, male breast cancer

## Abstract

**Simple Summary:**

In breast cancer oncology, great progress has been made towards a more personalised approach. In particular, the introduction of genomic signature testing has helped physicians select the best adjuvant treatment for hormone-receptor-positive, human epidermal growth factor receptor-2–negative early breast cancer. Although Oncotype DX is recognised worldwide as the preferred genomic test, there are still some areas of uncertainty and opportunity. The aim of this review is to discuss the most challenging and urgent issues related to its daily use, providing insights for better integration and wider application in clinical practice.

**Abstract:**

Several multigene assays have been developed to help clinicians in defining adjuvant treatment for patients with hormone-receptor-positive (HR+), human epidermal growth factor receptor-2 (HER2)–negative early breast cancer. Despite the 21-gene assay having been available for decades, it has only recently been included in the healthcare systems of several countries. Clinical optimisation of the test remains of critical interest to achieve a greater impact of genomic information in HR+/HER2- early breast cancer. Although current guidelines recommend the use of the 21-gene assay in early breast cancer at intermediate risk of relapse, the implication of the Recurrence Score (RS) in some grey areas still remains uncertain. Our aim is to critically discuss the role of RS in peculiar circumstances. In particular, we focus on the complex integration of genomic data with clinicopathological factors; the potential clinical impact of RS in node-positive premenopausal women and in the neoadjuvant setting; the significance of RS in special histologies and in male patients; and the management and time-optimisation of test ordering. In the absence of robust evidence in these areas, we provide perspectives for improving the use of the 21-gene assay in the decision-making process and guide adjuvant treatment decisions even in challenging cases.

## 1. Introduction

Clinical decision-making regarding the use of adjuvant treatment for invasive early breast cancer (BC) is largely based on a variety of factors, including anatomical stage, biological cancer features and patient characteristics. This information can be used to determine the individual risk of BC recurrence and to derive recommendations for optimal adjuvant treatment options. However, the treatment decision-making process remains challenging, especially in HR-positive (HR+)/HER2-negative (HER2-) BC at intermediate risk, where the additional benefit of chemotherapy (CT) over the use of endocrine therapy (ET) alone is largely unclear.

Prognostic factors for early-stage BC were derived from population-based data and were not able to accurately predict at the individual patient level. This inaccuracy led to a general prudential attitude that eventually led to an overuse of unnecessary adjuvant CT, with potentially avoidable iatrogenic toxicities. In this scenario, several genomic profiling tests have been developed and validated to help physicians select the optimal adjuvant systemic therapy for each individual patient. The recent update of the ASCO practice guidelines recommends the use of multigene assays (MGAs) in adjuvant decision-making for HR+/HER2- early BC and highlights the optimal placement of tests in different patient settings, recognizing the 21-gene test (Oncotype DX) as one of the most widely used [1]. In this narrative review, we have attempted to critically comment on the role of Oncotype DX in some critical grey areas where uncertainties and opportunities in the use of genomic tests are still a matter of debate.

## 2. Should We Integrate Standard Clinicopathological Risk Factors and Genomic Score?

While the available second-generation genomic assays (MammaPrint, EndoPredict, Prosigna, Breast Cancer Index) include some of the well-established clinicopathological (CP) factors in the automated algorithm for generating the genomic score, Oncotype DX does not and the Recurrence Score (RS) purely defines selected molecular characteristics regardless of the CP features [2]. 

It has been shown that the opportunity to integrate CP risk factors, including age, tumour size, nuclear grading, and hormone receptor levels, may refine the test’s accuracy [3]. Actually, clinical risk features alone (tumour size, grade, age) have been shown to be prognostic but not predictive of CT benefit [4]. However, integrating these CP factors with the genomic risk offers the potential for a more accurate prognosis that ultimately supports the choice of optimal adjuvant therapy.

A prespecified secondary analysis of TAILORx assessed the impact of clinical risk on prognosis. Clinical high-risk patients were classified as high and low risk according to the binary categorisation used in the MINDACT trial and Adjuvant! Online. Patients with identical RS but different clinical risk levels provided very different prognostic estimates. Overall, clinical high-risk patients presented a 2.5- to 3-fold higher risk of recurrence at 9 years than low-risk patients [5]. When further stratified by clinical risk, women with a RS of 21 to 25 had a similar absolute CT benefit regardless of their clinical risk, whereas, for patients with an RS of 16 to 20, a higher estimated CT benefit was found in those at high clinical risk (6.5 ± 4.9%) but not in those at low clinical risk (−0.2 ± 2.1%). This highlights the importance of integrating clinical and genomic risk to obtain more accurate prognostic results [5].

This opportunity has been implemented in the RS-pathology and clinical (RSPC) model, which integrates the genomic results with some clinical risk factors (age, tumour size, grade and type of ET) and ultimately derives a more robust early BC prognosis compared to the reduced model of either RS alone or CP risk factors alone. Notably, the RSPC model re-classified a not-negligible number of intermediate-risk patients by RS as low risk (64% vs. 54%), whereas the high-risk category did not change, except for some inconsistencies in the case of small and low-grade tumours [6]. 

In contrast to the RSPC model, which predicts prognosis but not CT benefit, the more recent RSClin tool adds both prognostic and predictive information to assist adjuvant treatment decisions by integrating CP risk factors into an algorithm generated from individual data analysis of three different clinical trials (NSABP B-14, NSABP B-20, and TAILORx) [7]. RSClin is expected to recalibrate the purely prognostic/predictive estimation of RS and provide crucial information to better support the decision-making process for adjuvant therapy in early BC without nodal involvement. Table 1 provides an example of how the use of RSClin may ultimately impact on physician choice and patient preference after refining prognosis and treatment prediction estimates from RS.

Based on the relevant accuracy refinement, we expect that calculators such as RSPC and RSClin will be appropriately implemented in the daily clinical practice in order to individualise prognosis and associated risk among women with HR+/HER2- early BC.

## 3. Should We Test pN1 Premenopausal Women?

Since about one-third of patients with newly diagnosed HR+/HER2- early BC have lymph-node-positive disease, which is associated with an increased risk of recurrence [8], the role of genomic tests in this cohort of patients is crucial. The extent to which RS can predict CT benefit in this group was investigated in the RxPONDER trial [9], which evaluated patients with one to three positive axillary lymph nodes (N1) and RS up to 25. The results clearly demonstrated that postmenopausal women with HR+ N1 early BC can safely forego CT, minimizing unnecessary cytotoxic treatment exposure. This is not the same for premenopausal women who achieved a greater invasive disease-free survival (iDFS) benefit with the addition of CT, regardless of the value of RS (range 0–25) and the number of lymph nodes involved (range 1–3). The relative improvement of 40% in iDFS and 42% in distant disease-free survival (DDFS) in premenopausal women receiving CT raises the long-standing controversy about the true drivers of CT benefit in this young patient population [10]. In particular, the question of whether CT benefit for premenopausal women is due to a direct cytotoxic effect in eradicating micrometastatic disease or rather to the indirect effect of chemotherapy-induced amenorrhea (CIA) and ovarian function suppression (OFS) still remains a question of debate [11,12]. 

CIA, which occurs in approximately 40% of women receiving CT, has been associated with an overall survival benefit (OS) in HR+/HER2- early BC [13], supporting the role of OFS especially in patients at higher risk of recurrence. Results from the Tamoxifen and Exemestane Trial (TEXT) and the Suppression of Ovarian Function Trial (SOFT) demonstrated a 5% improvement in recurrence rates for HR+/HER2- premenopausal patients treated with OFS in addition to standard ET, with up to a 10% absolute improvement in 8-year recurrence-free survival (RFS). Therefore, it is possible that the use of endocrine treatment strategies stronger than tamoxifen, such as OFS plus an aromatase inhibitor (AI), would have provided benefits comparable to CT [14,15,16]. Similar results were found in the NSABP B-30 trial, which showed improvements in both OS and DFS among premenopausal women who experienced CIA for at least six months after adjuvant CT [12]. 

Similarly, the TAILORx trial reported a clear CT advantage over ET in those young patients who obtained early menopause but not in the case of persisting premenopausal status. Accordingly, patients aged 46–50 years derived a greater benefit as compared to those aged <40 years, suggesting that CIA plays a more relevant role than the direct cytotoxic effect of CT in a condition of induced and established OFS, as expected in perimenopausal (46–50 years) but not in younger women [5]. 

Moreover, the RxPONDER trial described the CT benefit in premenopausal patients with low genomic risk (RS < 25) in whom the estrogen-related features are known to predominate over the proliferation modules. This observation, along with the fact that the higher the RS, the greater the CT benefit, suggests that ET plus OFS may be a valid therapeutic option for younger women with RS < 25, as ET would be the more appropriate treatment to knock out the estrogen-related module than CT, which would instead be preferred when the proliferation module predominates in higher RS [17].

Finally, the results of the pre-operative ADAPT and ADAPT-cycle trials seem to have proceeded in the same direction [18,19,20]. In particular, the investigators reported a significant increase in the ET-response (Ki67 < 10%) in premenopausal patients in both groups of RS (0–25 and ≥26) when OFS was added to ET alone. The analysis also showed that in the most controversial group of premenopausal patients younger than 40 years, ET-response with AI plus OFS was up to 85% in RS 0–25 and 60% in RS ≥ 26, supporting the hypothesis that we can safely omit CT in favour of OFS plus ET in the majority young, N1, early BC patients based on clinical and genomic risk along with pre-operative ET-response [20].

Randomised clinical trials specifically addressing the question of whether OFS in combination with ET may be an appropriate alternative to chemoendocrine therapy in premenopausal women at intermediate risk of relapse would be desirable. However, retrospective analyses of prospectively conducted clinical trials could also provide interesting information. In the last San Antonio Breast Cancer Symposium (SABCS), an analysis of patients in the SOFT trial showed that the Breast Cancer Index (BCI), a gene expression-based signature able to predict the risk of distant recurrence and the benefit of extended ET in early-stage HR+/HER2- BC, was able to identify women who would benefit from the addition of OFS and exemestane. The study also confirmed the prognostic value of the test, as women with higher BCI risk scores were more likely to experience disease recurrence. The significance of this finding is that for the first time a genomic test can identify those women who derived benefit from OFS as well as those who did not, which is extremely important given the potential impact of this more intensive treatment approach in young women [21].

Mirroring the controversies, the ASCO 2022 recommendations state that premenopausal women with node-positive tumours should not be tested with MGAs [1], while the ESMO guidelines suggest that genomic tests, along with other CP factors, may be used to support decisions about systemic treatment in intermediate-risk patients when these determinations are challenging [22]. 

## 4. Should We Test Patients for Neoadjuvant Treatment?

Potential neoadjuvant strategies in patients with HR+/HER2- early BC include neoadjuvant chemotherapy (NAC) and neoadjuvant endocrine therapy (NET), with the aim of downstaging inoperable tumours, offering more conservative surgery, and adapting the adjuvant phase of systemic therapy [23]. However, data predicting the efficacy of each approach are lacking, and the optimal pre-operative treatment strategy still remains a challenge [24]. Given their proven efficacy in adjuvant treatment, much recent interest has focused on the potential role of MGAs in the neoadjuvant setting in order to identify patients who will derive the greatest benefit from NAC or NET [25]. The association between Oncotype DX RS and pathological complete response (pCR) has been investigated in several studies and a recent meta-analysis confirmed that the pCR rate was significantly higher in patients with a high RS compared to patients with a low and intermediate RS (11% vs. 1% respectively), suggesting that RS is a potential predictive factor able to accurately select patients for NAC [26].

Less used than NAC, the role of NET and the issue of optimal patient selection for such an approach are still debated [25,27]. Recently, NET has been shown to be an effective and safe pre-operative treatment option for HR+/HER2- postmenopausal early BC, with outcomes comparable to NAC as for conservative surgery and clinical response rates [28,29].

Similarly, some clinical trials have investigated the role of Oncotype DX in the context of NET, finding that the lower RS, the more effective the clinical impact of NET [27,30]. A recent meta-analysis of eight available studies evaluating Oncotype DX and NET confirmed these findings, reporting a lower clinical response rate in patients with high genomic risk (RS > 30) and a higher rate of partial response in patients with low (<11) and intermediate RS (RS 25–30), although the absolute number of patients achieving the pCR rate was limited [31].

Recently, a small number of pilot trials have been conducted to test the feasibility of using the RS as a decision aid for NET or NAC. Two phase II studies investigated the use of Oncotype DX in the initial diagnostic biopsy of HR+/HER2- early BC patients. Those with RS < 11 received NET, those with RS > 25 were treated with NAC, while patients in the intermediate group (RS 11–25) were randomised to receive either NET or NAC. As expected, no patient in the lower RS group and in the intermediate group treated with NET achieved pCR, whereas 22% of patients receiving NAC in the RS >25 group achieved pCR [32,33].

As mentioned above, beyond the clinical response rate, the biological impact of a short pre-operative NET in combination with genomic RS was investigated in the ADAPT trial, demonstrating the ability to refine the prognostic and predictive role of the RS incorporating CP factors and ET response [19,20]. 

## 5. Should We Test Patients with Histologies Other Than Ductal Carcinoma?

Invasive lobular carcinoma (ILC) is the second most common BC subtype after invasive ductal carcinoma (IDC), accounting for approximately 10–15% of all BC diagnoses [34]. ILC represents a different pathologic entity than ILD [35,36,37]. First, most ILCs have higher expression of ER and PgR compared to IDCs (90% vs. 70%), and few ILC cases are HER-2 amplified [35]. From a biological perspective, ILC is characterised by loss of E-cadherin and high expression of critical genes such as CDH1, ERBB, PIK3, and others within the TGF-B signalling pathway [38]. These molecular differences are thought to be responsible for the lower sensitivity to CT and for the lower overall survival rate as compared to IDCs [39]. 

Despite these differences, IDC and ILC are typically managed similarly. Consequently, the use of MGAs in ILC is applied in the same way as in IDC, although little data has prospectively investigated the role of MGA in ILC and in other special histologic subtypes. Conversely, several retrospective studies have addressed this issue and questioned the relevance of MGA, and RS in particular, as a prognostic or predictive factors in patients with ILC [39,40,41,42]. It has been shown that the prevalence of a high RS in patients with ILC is limited and the CT benefit expected in case of high RS is less pronounced as compared to those IDC with similar genomic scores [43,44,45]. These findings simply justify the general attitude of clinicians to consider RS less informative in the case of ILC patients.

A recent study by Weiser et al. examined the use of RS in a large group of 15,763 patients with IDC and ILC and found that the distribution of RS differed between the two subtypes, with fewer ILC patients having a high RS compared to IDC (6.6% vs. 16.0%). Although patients with ILC were less frequently treated with CT than patients with IDC (17% vs. 24.6%), the authors noted that RS retained its prognostic and predictive value in ILC as a clear CT benefit was found in case of high RS but not in the low to intermediate RS [46]. The importance of testing rarer histologies of BC is supported by several retrospective studies that showed correlations with RS. Histologies with a more favourable prognosis more often have low to intermediate RS, such as mucinous BC [47], while pleomorphic and anaplastic variants predominantly have intermediate to high RS [48]. 

Accordingly, international guidelines do not distinguish between different BC histologies when it comes to multigene testing. Further research is needed to thoroughly investigate the genomic heterogeneity of ILC and support clinicians in the treatment decision-making for patients with ILC.

## 6. Should We Test Male Patients?

Male breast cancer (MBC) patients were excluded from clinical trials that led to the development and validation of Oncotype DX. However, given the meaningful impact of the 21-gene signature in women and the overwhelming prevalence of HR+/HER2- subtype in MBC, which accounts for approximately 90% of all cases, understanding the role of genomic testing in this population is not trivial. 

Over the past decade, several retrospective studies have evaluated the use of Oncotype DX in MBC using large population-based datasets. A recent systematic review and meta-analysis of six retrospective cohort studies compared the RS in male and female BC patients and found that MBC has more advanced tumour stages and higher grades than females. Although higher RS was more common in men, no gender-specific pattern of RS distribution was observed at the meta-analysis level, with no significant difference between women and men in all RS categories [49].

In contrast, two different studies showed a well-defined different pattern of RS distribution in men [50,51]. While the average RS was similar in men and women, men had a significantly higher risk than women of falling into the high RS (RS ≥ 31: 12.4% vs. 7.4%, respectively) and low RS categories (RS ≤ 10: 33.8% vs. 22.1%). Notably, both trials supported the prognostic value of RS in men: the higher the RS category, the lower the survival rate [50]. Moreover, RS was associated with an increased risk of death in men at a much lower threshold than in women. In men, the risk of death increased until RS 21, after which the risk plateaued. Finally, in contrast to their female counterpart, male patients with intermediate RS had a much higher mortality risk than those with low RS [51].

Although heterogeneous and somewhat conflicting, these data highlight the need for dedicated research to prospectively validate the 21-gene signature in the MBC population. Different underlying mechanisms suggest a gender-specific pathogenetic background as a different distribution of somatic genetic alterations has been described (e.g., a lower frequency of PI3KCA and TP53 mutations and a higher rate of DNA repair-related genes mutation) [52]. Indeed, quantitative analysis of the 21-gene expression showed that men have a meaningful higher representation in estrogen-related genes as well as in both proliferation and invasion pathways compared to women. Furthermore, the prevalence of BRCA1/2 germline mutations is notable in MBC, accounting for about 15–20% in HR+/HER2- disease [53]. Finally, crosstalk between androgen and estrogen pathways has been suggested to influence the ET response to some extent, without clarifying the clinical implications [54]. Whether and how all these unique molecular features might translate into a different clinical approach is still unknown.

The studies by Wang and Massarweh question the appropriateness of the traditional female RS cut-offs in capturing the different prognoses of MBC according to their RS [50,51]. In addition, there is a lack of data, either prospective or retrospective, on the predictive value of RS in men, as the impact of this treatment on patients’ outcomes is not reported. The need for a more tailored MBC approach strikes with the extreme effort in reaching an adequate sample size to perform randomised trials. In this perspective, a collaborative network, the International Male BC Program, was launched by the joining efforts of several worldwide cancer research organisations in the hope of addressing this unmet need.

## 7. Should We Optimise the Timing of the Test?

Timing between surgery and adjuvant CT is crucial for improved survival as delaying the start of adjuvant CT can negatively affect long-term outcomes [55].

According to Farolfi, each week of delay in adjuvant CT initiation in rapidly proliferating early BC patients resulted in an HR of 1.01 (95% CI, 1.00–1.02; *p* = 0.067) for DFS and an HR of 1.01 (95% CI, 1.00–1.02; *p* = 0.124) for OS [56]. An Italian study examining the quality of care in seven different regions found, among other indicators, that 64.5 to 75.1% of patients start adjuvant treatment (either CT or ET) within 60 days after surgery [57]. In this context, the extra time needed to obtain the genomic score is a hot topic.

Usually, all new cases of early BC patients are discussed in a multidisciplinary team (MDT), where physicians may request Oncotype DX according to the established local criteria for test access. It then takes 10 to 16 days for the available results, leaving little time for physicians to inform patients and choose the most appropriate treatment strategy depending on the genomic score. Since the time from surgery to test result is approximately 26 days, efforts have been made to shorten this time as much as possible. In this context, one option would be to develop an early, standardised ordering criterion based on established CP features and to shift the protagonist of the Oncotype DX query from the medical oncologist to the pathologist, who will request the test once the final report is available. The identification of CP features that recognise patients likely to benefit from RS should be trivial and standardised and based on the general attitude to request the test when these CP features are met. In this scenario, a Korean retrospective study developed and validated a nomogram based on three simple parameters, nuclear grade, PgR expression and Ki67 to predict the RS [58]. Tumour grade and HR status also proved to be strong predictive factors in the development of other nomograms for RS anticipation [59].

Furthermore, some surgical centres evaluated the clinical use of Oncotype DX “reflex” test for ordering automatically the test if certain CP features are encountered. The test criteria included a combination of tumour size, lymph node involvement, nuclear grade, and age and support a timely and immediate test ordering once the pathology report is completed and appropriate [60]. Using this approach, Losk et al. have shown a crucial time-saving effect, by reducing the number of days (6.5) between surgery and the start of CT based on the RS result [61]. 

As waiting time becomes an increasingly important concern in oncology, breast units could consider in selected cases the “reflex” test approach to reduce the detrimental delay in starting the adjuvant treatment.

Moreover, facilitating patient access to Oncotype DX could reduce disparities, increase test appropriateness, and greatly improve patient satisfaction and consistency of patient care (Figure 1).

## 8. Conclusions

Oncotype DX is the only test that provides both prognostic and predictive information of CT benefit [4] and is a crucial part of the decision-making process for early luminal BC [1]. The test is not a black-and-white answer, but a dynamic instrument that should be adapted to meet clinical demands.

Further research should address these grey areas, which we have discussed in the present review. As genomic testing has an independent prognostic role, integrating it with CP characteristics through innovative tools is essential to tailor treatment [7]. Premenopausal women with node-positive disease should be tested; however, future studies should focus on assessing whether the relative benefit of CT is due to an indirect endocrine effect or eradication of micrometastatic disease. The San Gallen Consensus Conference endorsed the use of MGAs as a strategy for an optimal neoadjuvant approach, favouring NET in case of low genomic signature and NAC in case of high-risk genomic score [62,63]. However, due to a lack of phase 3 trials, ASCO guidelines discourage this approach [64]. Little is known about IDC histologies, which appear to have distinct RS based on their intrinsic aggressiveness [48]. Sex-specific risk thresholds should be established through inclusion of male patients in clinical trials. Finally, strategies are urgently needed to avoid harmful CT delays, influenced by the timing of requests for genomic testing [60,61].

Clinical optimisation of the test continues to be of utmost importance in order to achieve a greater impact of genomic information in HR+/HER2- early BC, even in peculiar and challenging circumstances. 

## Figures and Tables

**Figure 1 cancers-15-00148-f001:**
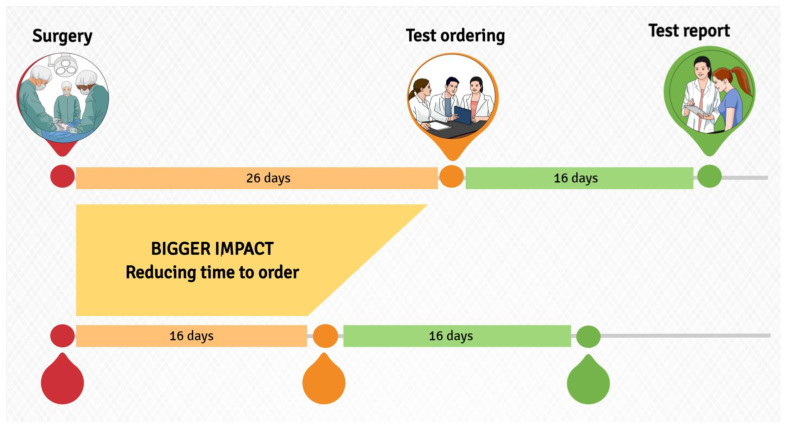
Expected benefit by improving workflow and streamline processes.

**Table 1 cancers-15-00148-t001:** Example of how the integration of RS Clin can refine the indication for adjuvant treatment, especially for intermediate to high RS value.

Patient of 40 yo, N0, 1 cm, Grade 1				
		RS 24	RS 27	Delta CT
**RS alone**	9y DR ET	10%	16%	
	CT benefit	<1%	>15%	**15%**
**RS + CP (low)**	9y DR ET	11.5%	16%	
	CT benefit	6.5%	>15%	**8.5%**
**RS Clin**	10y DR TAM	7%	7%	
	CT benefit	3%	4%	**1%**

An RS of 24 corresponds to a 9-year DR risk of 10%, which increases to 16% with RS of 27. The prediction of benefit from CT changes significantly when the threshold of 25 is exceeded: for an RS of 24, the benefit of CT is less than 1%, which increases to more than 15% for an RS of 27, with a delta of 15% for a difference of only 3 points in the RS value. After adjustment for RS Clin, which aims to integrate genomic features and CP characteristics, DR at 10 years is 7% for both RS 24 and 27. The benefit of CT with the implementation of this tool decreases to only 3% and 4%, respectively, nullifying the previously described delta. Thus, integrating RS Clin identifies high-risk patients and ensures that a potentially toxic treatment is administered only to those patients who will truly benefit. *(Adapted from “View from the Trenches: What Will You Do on Monday Morning?” SABCS 2020, M. Regan).*
**Abbreviations**: CP: clinicopathological factors; CT: chemotherapy; DR: distant recurrence; TAM: tamoxifen.

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
