# Peer review of "Controversies and Opportunities in the Clinical Daily Use of the 21-Gene Assay for Prognostication and Prediction of Chemotherapy Benefit in HR+/HER2- Early Breast Cancer"

_cancers, 2022, doi:10.3390/cancers15010148_

Round 1

Reviewer 1 Report

The authors present a review on the use of genomic tests for therapy guidance in early breast cancer HR+/HER2-. They address several questions aiding in structuring the article. As they describe themselves, it is a narrative review, bearing the challenge for the reader that some clarity is lacking and the interest of the reader is lost during reading. This is especially true for the discussion which should be shortened and re-structured.

Author Response

We are grateful that you appreciate our manuscript. The idea of structuring the paper to address some of the key unanswered questions in clinical practice regarding the use of Oncotype DX was a means of keeping the reader's attention high and providing some points for reflection. We particularly appreciated your suggestion to shorten the discussion. We have tried to restructure the conclusion to give a better overview of what the aim of our work was and how it can help physicians in clinical practice.

Changes are marked in “track changes”. We hope that our efforts will lead to the acceptance of the revised manuscript.

Reviewer 2 Report

The investigators have clearly outlined what might be considered the price of success.  In 2000 a consensus conference identified the over-treatment of early stage, node-negative primary tumors as a major clinical issue in breast cancer (JNCI 93:979, 2001). There was a strong suspicion that a useful clinical classifier would at a minimum combine indicators of cell proliferation with the established hormonal biomarkers, but approaches based on immunohistochemistry were not up to the task. Multiplex PCR by that point had matured technically so that a panel of analytes of moderate size could be measured reproducibly on formalin-fixed paraffin embedded tissue. The developers of OncoType DX presented a straightforward algorithm to generate a risk score and published retrospective data compelling enough to justify two randomized phase III trials. In 2022 the updated results of TailoRx show that the primary conclusion still holds, and that thousands of women can safely forgo the burdens and risks of chemotherapy (Sparano et al, San Antonio Breast Cancer Conference). The RxPONDER trial extended these results to post-menopausal patients with a low burden of nodal disease (cited by the authors).

How much further can we go? The authors of this paper turn their attention to groups of patients who were either not included in TailoRx or RxPONDER, or for whom the results of these trials were less clear cut. Does the OncoType RS remain predictive when the biology of either the treatment or the disease itself differs from the setting of chemotherapy for post-menopausal invasive ductal carcinoma? 

Even the simplest version of this question is strikingly hard to answer. Does it matter whether the chemotherapy is delivered before or after surgery? The patients offered neoadjuvant treatment typically differ from those enrolled in TailoRx and RxPONDER; they often have larger tumors. The trials have been small and they used pCR rather than recurrence to assess benefit. Do some of them include a recurrence objective, so that with maturing data we might see the effect?

For pre-menopausal patients the question is whether the benefits of chemotherapy derive from the eradication of residual disease or from the suppression of ovarian function. Even after several large, well-designed trials the question remains open. Without this understanding it is difficult to interpret the RS for these patients.

The authors consider patients with forms of breast cancer other than ductal carcinoma. Here there is probably reason for more caution than these authors recommend. In their discussion of premenopausal patients the authors note that the RS interrogates the complex interaction between estrogen response and cell proliferation in ductal carcinomas. In female lobular and mucinous breast cancer subtypes influences of the tumor microenvironment may overpower the tumor features to which the OncoType RS is tuned, and in male breast cancer the hormonal stimuli are very different. The fact that the distribution of RS values in these populations differs from that in ductal carcinoma hints that these differences may be important. Notably a prognostic classifier developed by the same company for DCIS underperforms clinical factors even after several rounds of refinement (Ann. Surg. Oncol. 26:3282, 2019). 

Parenthetically, that discussion of hormone response vs cell proliferation in the section on premenopausal cancer is the one part of the paper where I would recommend a clarifying re-write.  For a reader unfamiliar with how the RS algorithm was constructed it may be difficult to understand how the "ER-gene module" was "thresholded," and of course there is no "estrogen gene."

My final question for the authors is how best to approach these uncertainties. It is unlikely that randomized trials on the scale of TailoRx and RxPonder will be performed to address them. Can the authors suggest how observational studies could be designed to provide useful if not definitive guidance?

Author Response

We would like to express our sincere gratitude for the thorough analysis of our manuscript and the suggestions provided by Reviewer 2.  We are grateful that you appreciate the relevance and appropriateness of our work.

We agree with you that the “proliferation module” in the premenopausal paragraph was not clear enough for a reader who is not familiar with Oncotype DX. We have therefore rewritten this part and hope that it is now clearer.
We particularly appreciated your last comment about how we would deal with these uncertainties. We agree that it would be quite difficult to conduct randomized clinical trials even though it would be the preferred option. We also believe that retrospective observation of prospectively conducted trials could provide some interesting information. As an example of this, we cited the analysis of the Breast Cancer Index in the SOFT trial presented at the last SABCS 2022. Since Reviewer 1 suggested shortening the conclusion to provide only the most relevant information, we add this comment in the premenopausal paragraph.

Please refer to the final version of the manuscript where we have included the recommended changes that are marked in “track changes”. 
